# Burning Velocities of Pyrotechnic Compositions: Effects of Composition and Granulometry

Charles Rosères [1,2], Léo Courty [1,*], Philippe Gillard [1] and Christophe Boulnois [2]

1    Univ. Orleans, INSA-CVL, PRISME EA 4229, 63 Avenue de Lattre de Tassigny,
     CEDEX, 18020 Bourges, France; charles.roseres@orange.fr (C.R.); philippe.gillard@univ-orleans.fr (P.G.)
2    Nexter Munitions, Energetic Materials and Pyrotechnics Department, 7 Route de Guerry,
     CEDEX, 18023 Bourges, France; c.boulnois@nexter-group.fr
*    Correspondence: leo.courty@univ-orleans.fr

**Abstract:** Burning velocities of binary and ternary pyrotechnic compositions are measured in gutter. The study focuses on the determination of the joint influence of several parameters: oxidant/reducer ratio, reducer granulometry, and binder content. Measurements are performed following the standard NF T70-541 for burning velocity estimation using an optical acquisition method. Binder content has a linear influence on the burning velocity with a pivot point in slope at supposed stoichiometry. Changing the granulometric class of metallic reducer shows to have different influences before and beyond a 20% diameter reduction.

**Keywords:** energetic materials; pyrotechnic compositions; burning velocity in gutter; DNAN; Mg; $SrO_2$

## 1. Introduction

Pyrotechnic compositions are energetic materials used for the creation of light, of gases, or for heat production. They have multiple applications and are used for both civilians and military applications. This wide range of applications leads to the use of many kinds of chemical compounds for their formulations. Some of these compounds can be harmful for living beings or the environment and can be forbidden by the REACh regulations. Therefore, it can be necessary to replace some of these materials. To do so, a deep knowledge and mastery of the different involved phenomena is necessary. Multiple parameters have an influence regarding burning velocities [1]. However, there are in the literature very few models concerning pyrotechnics composition, compared to explosives or propellants. We can cite, for instance, the Emmons problem (1956) for the combustion of propellant subject to transfer flow [2]. More recently, a moving boundary modeling approach was developed for solid propellant combustion [3].

The wide range of parameters influencing burning velocity of pyrotechnic compositions makes difficult the estimation of the influence of each of them. Some of these parameters have been studied in the literature: powder granulometry or composition of the couple oxidant-reducer. Nevertheless, their fine influence or their joint influence are not deeply studied [1,4,5]. For the powder granulometry, we can cite the work of Weiser et al. on the ignition of energetic materials for different metal particle size [5]. They found that too fine of particles are oxidized too fast and make the ignition delay longer. Best results would be obtained for a mixture of coarse and ultra-fine particles. For the composition of pyrotechnic mixtures (couple oxidant-reducer), we can cite the very recent work of Biegańska and Barański on pyrotechnic compositions producing an acoustic effect [6].

This paper aims at studying the joint influences of several parameters on the burning velocities of model pyrotechnic compositions: composition (including binder content), granulometry, and porosity.

Studied pyrotechnics compositions are composed of an oxidant, strontium peroxide $SrO_2$, and two reducers: a metal reducer, magnesium Mg, and an energetic molecule 2,4-dinitroanisole (DNAN) used as mechanical binder.

The main studied parameters are the reducer particle size (magnesium granulometry), the oxidant/reducer ratio, and the binder content. Present study focuses on binary and ternary compositions composed of the different materials cited above. Let us notice that $SrO_2$ is rather common in the pyrotechnics field, but thermodynamic data are little known; on the contrary, DNAN has been more studied [7,8].

The influence on burning velocity of reducer/oxidant content, magnesium particle size, porosity, and binder content is investigated on 60 pyrotechnic compositions. Thermal analyses experiments have already been conducted on selected compositions [9]. The burning velocity is measured using a "U" shaped gutter following the standard NF T70-541 [10].

The aim of this work is to study the influence of the cited parameters for binary and ternary pyrotechnic compositions on burning velocities to optimize future formulations.

The following section presents the materials and methods, as well as the experimental setup. Results and discussions are presented in Sections 3 and 4.

## 2. Materials and Methods

### 2.1. Materials

The pyrotechnic compositions studied in the present work were manufactured using three raw components: magnesium (Mg), strontium peroxide ($SrO_2$), and DNAN. The manufacturing process is the sole intellectual property of Nexter Munitions and cannot be reproduced here. Pyrotechnic compositions grains were granulated using DNAN as a binder.

The raw components were described in a previous article [9] regarding their granulometry as well as their thermal behaviour. Concerning the magnesium powder used for ternary compositions, three granular classes A, B, and C were used:

1. $140 \ \mu m < A < 200 \ \mu m$
2. $200 \ \mu m < B < 250 \ \mu m$
3. $250 \ \mu m < C < 300 \ \mu m$

Two binder contents were studied: 5 wt.% and 10 wt.% of DNAN, and oxidant/reducer (ox/red, $SrO_2$/Mg) ratios were ranging from 50/50 to 90/10 in wt.% for ternary compositions. Two types of binary compositions were studied: $SrO_2$/DNAN and Mg/DNAN with wt.% varying, respectively, from 95/5 to 50/50 and from 90/10 to 40/60.

Three main chemical reactions are supposed to occur: $SrO_2$/Mg, $SrO_2$/DNAN, and Mg/DNAN, but only two kinds of binary compositions were studied. To limit the influence of the manufacturing process and since the reaction between $SrO_2$ and Mg is supposed to be the main chemical reaction, only the compositions $SrO_2$/DNAN and Mg/DNAN were made to study their likeness to occur, undergoing the same manufacturing process. Table 1 presents the formulations of the studied binary compositions.

**Table 1.** Binary compositions formulation.

| Composition | SrO$_2$/DNAN Reaction | | Mg/DNAN Reaction | |
|:---:|:---:|:---:|:---:|:---:|
| | wt.% SrO$_2$ | wt.% DNAN | wt.% Mg | wt.% DNAN |
| a | 95 | 5 | 90 | 10 |
| b | 90 | 10 | 80 | 20 |
| c | 85 | 15 | 70 | 30 |
| d | 80 | 20 | 60 | 40 |
| e | 70 | 30 | 50 | 50 |
| f | 50 | 50 | 40 | 60 |
| g | 87.88 | 12.12 | - | - |

Composition *g* corresponds to the stoichiometric reaction between $SrO_2$ and DNAN when $CO_2$ is supposed to be the main combustion product for DNAN:

$$12\,SrO_2 + C_7H_6O_5N_{2\,(DNAN)} \Rightarrow 12\,SrO + 7\,CO_2 + 3\,H_2O + N_2 \tag{1}$$

Mg is supposed to react with DNAN at high temperatures.

Ternary compositions were formulated to express the influence of ox/red ratio, magnesium granulometry, and binder content on burning velocity, undergoing the same manufacturing process. Table 2 presents the formulation of the studied ternary compositions, along with the ox/red ratio.

**Table 2.** Ternary compositions formulation with ox/red ratios.

| Composition | wt.% Mg | wt.% SrO$_2$ | wt.% DNAN | ox/Red Ratio (%) |
|:---:|:---:|:---:|:---:|:---:|
| X1 | 47.5 | 47.5 | 5 | 50 |
| X2 | 34.2 | 60.8 | 5 | 64 |
| X3 | 32.3 | 62.7 | 5 | 66 |
| X4 | 28.5 | 66.5 | 5 | 70 |
| X5 | 23.75 | 71.25 | 5 | 75 |
| X6 | 19 | 76 | 5 | 80 |
| X7 | 14.25 | 80.75 | 5 | 85 |
| X8 | 13.49 | 81.51 | 5 | 85.8 |
| X9 | 11.4 | 83.6 | 5 | 88 |
| X10 | 9.5 | 85.5 | 5 | 90 |
| X11 | 45 | 45 | 10 | 50 |
| X12 | 32.4 | 57.6 | 10 | 64 |
| X13 | 30.6 | 59.4 | 10 | 66 |
| X14 | 27 | 63 | 10 | 70 |
| X15 | 22.5 | 67.5 | 10 | 75 |
| X16 | 18 | 72 | 10 | 80 |
| X17 | 13.5 | 76.5 | 10 | 85 |
| X18 | 10.8 | 79.2 | 10 | 88.78 |
| X19 | 10.1 | 79.9 | 10 | 88 |
| X20 | 9 | 81 | 10 | 90 |

Three kinds of ternary compositions were studied depending on the magnesium granulometry class (A, B, or C) used during the manufacturing process. The corresponding composition will be referred accordingly, e.g., A12 for the composition X12 in Table 2, manufactured with magnesium of granular class A. In total, 60 compositions were manufactured.

### 2.2. Methods

Experimental set up is described in the AFNOR standard NF T70-541 [10] regarding the measurement of burning velocity in small gutter. It is made of a parallelepipedal shaped bar made of stainless steel, 500 mm long and 20 mm wide, containing a gutter of width and depth of 10 mm, in which the pyrotechnic composition is placed. The burning velocity is acquired at a 300 mm distance using an acquisition device. About 25 g of composition is placed in the gutter for each experiment. The volume of the gutter is of $3.2 \times 10^{-5}$ m$^3$.

In the present study, choice has been made to measure the burning velocity using OKE 1000B optical fibres with four acquisition points, instead of the two points indicated in the standard (Figure 1). A YOKOGAWA DLM 2054 oscilloscope paired with an optronic converter is used as the acquisition device. Ignition of the pyrotechnic composition is conducted using a heated tungsten wire. Two trials were performed for each binary composition.

Compositions are placed in bulk in the gutter, and a level is then equalised along the gutter. Therefore, volume is controlled but not editable.

For the four acquisition points (C1–C4), the time corresponding to 50% of relative intensity is extracted. An example of the obtained raw data is presented in Figure 2. Burning velocities are then obtained by a linear regression on the plots distance as a function of time.

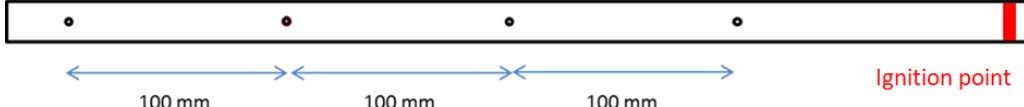

**Figure 1.** Scheme of the experimental gutter.

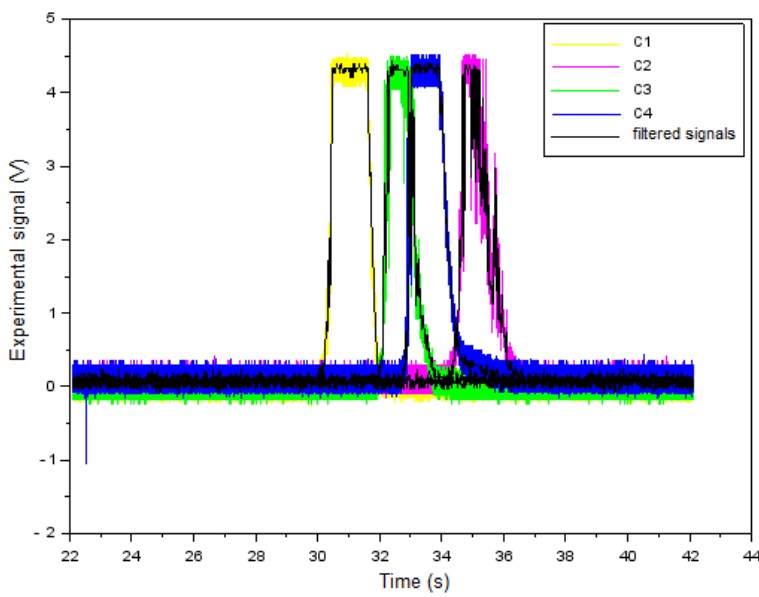

**Figure 2.** Example of the obtained raw data from optical fibre.

The aim of this article is to present experimental results on model pyrotechnic compositions. No statistical methods are used to do so, a range for each parameter is initially chosen (presented previously in this section). Non-statistical analysis of trends is only possible, and results are presented in the next section.

## 3. Results

### 3.1. Binary Compositions

Two kinds of binary compositions were studied, $SrO_2$/DNAN and Mg/DNAN. Mg/DNAN compositions could not be ignited. Burning velocities $v$ measured for $SrO_2$/DNAN compositions are presented in Table 3, as well as used experimental masses $m_{exp}$ and calculated porosity $\varepsilon$.

**Table 3.** Measured binary compositions burning velocities with associated experimental masses and porosities.

| Composition | Trial 1 | | | Trial 2 | | |
|---|---|---|---|---|---|---|
| | $m_{exp}$ (g) | $\varepsilon$ | $v$ (cm·s$^{-1}$) | $m_{exp}$ (g) | $\varepsilon$ | $v$ (cm·s$^{-1}$) |
| a | 31.71 | 0.75 | 0.88 | 31.26 | 0.75 | 0.63 |
| b | 27.79 | 0.76 | 3.21 | 26.99 | 0.76 | 3.42 |
| c | 25.85 | 0.75 | 2.62 | 24.86 | 0.76 | - |
| d | 24.99 | 0.74 | 2.03 | 23.60 | 0.76 | 1.77 |
| e | 22.83 | 0.73 | 1.37 | 22.88 | 0.73 | 1.11 |
| f | 19.28 | 0.71 | 0.2 | 19.07 | 0.71 | 0.26 |
| g | 27.93 | 0.75 | 3.06 | 27.07 | 0.75 | 3.34 |

Porosity $\varepsilon$ was calculated using the true density $\rho_{th}$ and apparent density $\rho_{(app)}$:

$$\varepsilon = 1 - \left( \frac{\rho_{app}}{\rho_{th}} \right) \qquad (2)$$

Figure 3 presents burning velocities of SrO$_2$/DNAN compositions regarding trials 1 and 2.

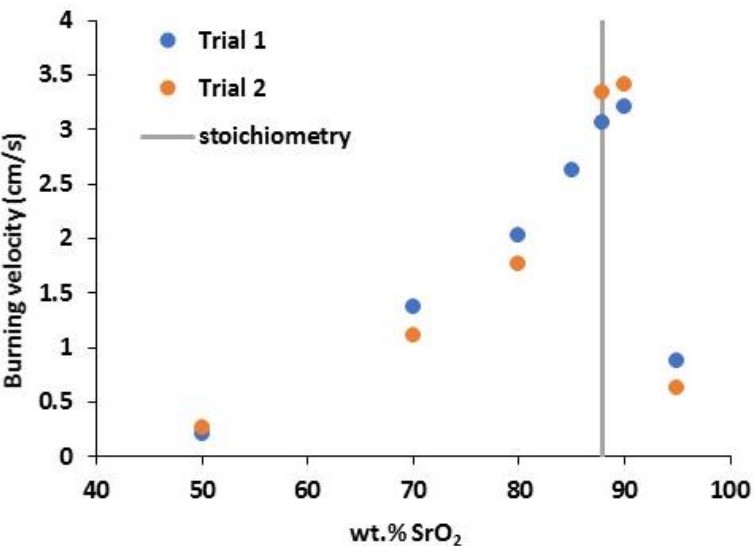

**Figure 3.** Measured burning velocities of binary SrO$_2$/DNAN compositions as functions of oxidiser content.

### 3.2. Ternary Compositions

The measured burning velocities of ternary compositions are presented in Figure 4 for the three Mg granulometric classes studied. The burning velocities and porosity values are presented in Table 4.

**Table 4.** Measured ternary compositions burning velocities and associated porosities.

| Composition | A | | B | | C | |
|---|---|---|---|---|---|---|
| | V (cm·s$^{-1}$) | Porosity $\varepsilon$ | V (cm·s$^{-1}$) | Porosity $\varepsilon$ | V (cm·s$^{-1}$) | Porosity $\varepsilon$ |
| X1 | 12.5 | 0.59 | 9.69 | 0.58 | 6.93 | 0.58 |
| X2 | 13.6 | 0.58 | 10.71 | 0.65 | 10.83 | 0.62 |
| X3 | 15.29 | 0.64 | 11.84 | 0.63 | 9.47 | 0.65 |
| X4 | 16.82 | 0.64 | 11.71 | 0.61 | 8.59 | 0.61 |
| X5 | 12.69 | 0.64 | 12.67 | 0.66 | 7.25 | 0.63 |
| X6 | 14.5 | 0.65 | 7.28 | 0.66 | 6.06 | 0.62 |
| X7 | 12.54 | 0.68 | 5.49 | 0.68 | 5.42 | 0.66 |
| X8 | 10.89 | 0.64 | 6.18 | 0.67 | 4.49 | 0.67 |
| X9 | 8.25 | 0.67 | 4.99 | 0.71 | 3.92 | 0.64 |
| X10 | 6.94 | 0.67 | 4.78 | 0.70 | 3.12 | 0.71 |
| X11 | 6.59 | 0.57 | 4.73 | 0.58 | 4.74 | 0.57 |
| X12 | 6.98 | 0.63 | 6.65 | 0.59 | 5.5 | 0.63 |
| X13 | 9.55 | 0.64 | 6.83 | 0.62 | 6.07 | 0.63 |
| X14 | 10.56 | 0.65 | 4.82 | 0.63 | 5.86 | 0.62 |
| X15 | 9.97 | 0.64 | 5.33 | 0.64 | 5.24 | 0.63 |
| X16 | 9.9 | 0.67 | 4.72 | 0.63 | 4.5 | 0.63 |
| X17 | 5.51 | 0.66 | 3.73 | 0.65 | 4.39 | 0.65 |
| X18 | 7.84 | 0.67 | 3.68 | 0.66 | 3.98 | 0.67 |
| X19 | 6.27 | 0.66 | 4.04 | 0.66 | 4.11 | 0.67 |
| X20 | 7.13 | 0.67 | 3.63 | 0.66 | 4.27 | 0.67 |

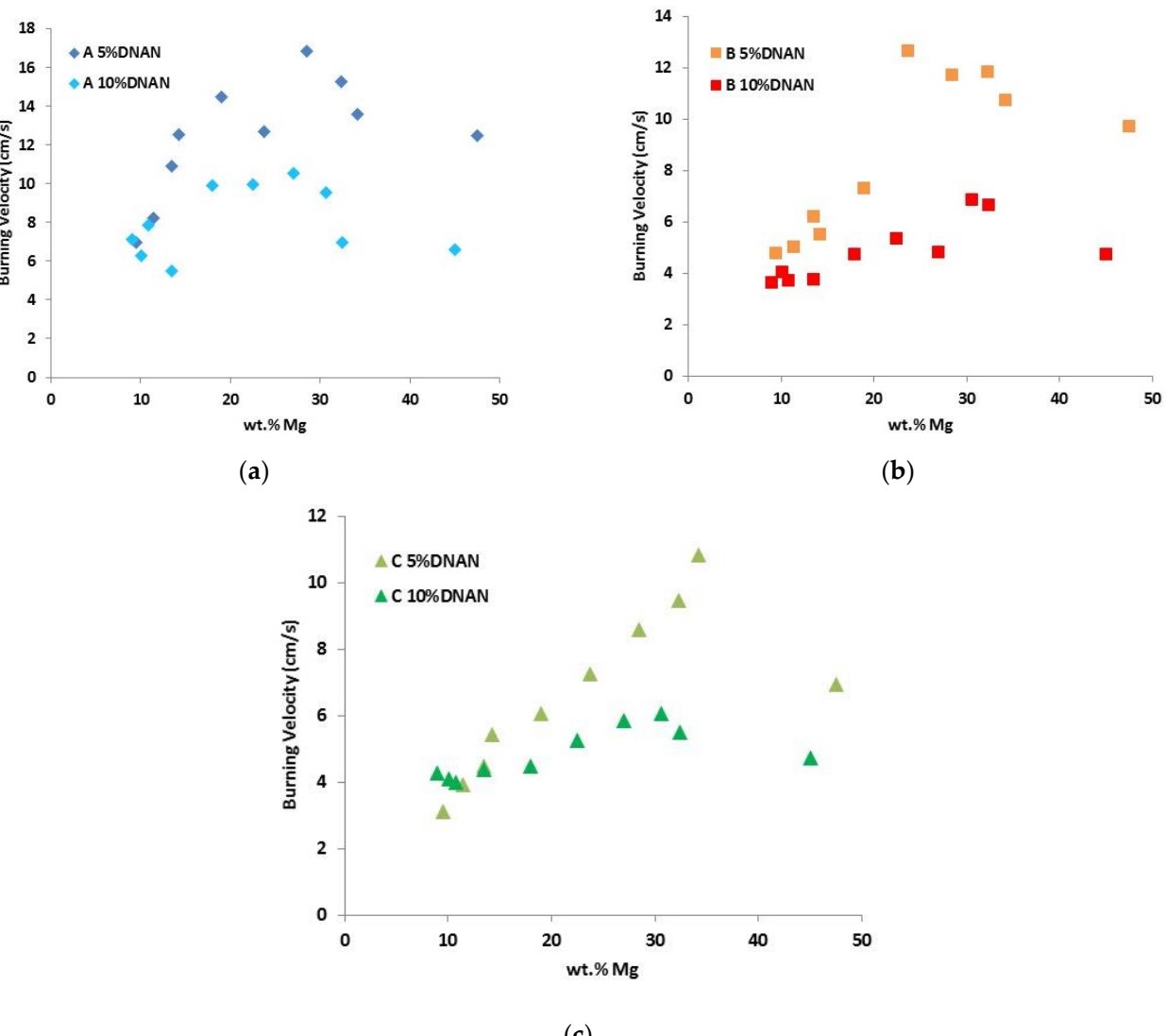

**Figure 4.** Measured burning velocities of ternary $SrO_2$/Mg/DNAN compositions: (**a**) Compositions using Mg of class A; (**b**) Compositions using Mg of class B; (**c**) Compositions using Mg of class C.

For each magnesium granulometric class, the stoichiometric compositions are X8 and X18 and will be referred to as such. The supposed chemical reaction to take place is presented in Equation (3).

$$(5 + x)SrO_2 + xMg + C_7H_6O_5N_2 \rightarrow (5 + x)SrO + xMgO + 7CO + 3H_2O + N_2 \quad (3)$$

## 4. Discussions

### 4.1. Binary Compositions

Composition C in trial 2 could not be ignited. The measured burning velocities are ranging from 0.2 $cm \cdot s^{-1}$ for composition F to 3.42 $cm \cdot s^{-1}$ for composition B. Maximum velocity is reached for a 90/10 $SrO_2$/DNAN composition. This formulation is close to the stoichiometric one at 87.88/12.12 (G), confirming the chemical reaction stated in Equation (1). As a reaction between Mg and DNAN is unlikely to happen, it appears that a reaction between $SrO_2$ and DNAN occurred and can play a role in the reaction between Mg and $SrO_2$ in ternary compositions.

### 4.2. Ternary Compositions

#### 4.2.1. Ox/Red Ratio Influence

It has been shown many times that ox/red content has an influence on the burning velocity of pyrotechnic compositions [1,11–14]. Increasing the reducer content tends to increase the burning velocity up to a certain point. Maximum burning velocity is often found at a higher reducer content than stoichiometry [1]. The studied pyrotechnic compositions follow the same rule, as it can be seen in Figure 4. Burning velocity increases from low Mg contents until a maximum, around 30 wt.% Mg, then it decreases until 50 wt.% Mg. It is supposed that the burning velocity would continue to decrease with higher Mg content. It is observed that the maximum velocity is obtained at much higher reducer content in contrast to what was noticed for binary compositions.

The maximum velocity obtained for compositions A is 16.82 cm·s$^{-1}$ for A4 at 28.5 wt.% Mg and is 10.56 cm·s$^{-1}$ for A14 at 27 wt.% Mg. Same observation is made for compositions B with 12.67 cm·s$^{-1}$ and 6.83 cm·s$^{-1}$ at 23.75 wt.% Mg and 30.6 wt.% Mg, respectively (B5, B13). For compositions C, maximal velocities are 10.83 cm·s$^{-1}$ and 6.07 cm·s$^{-1}$ at 34.2 wt.% Mg and 30.6 wt.% Mg, respectively (C2, C13). These comments are made for the two contents of DNAN studied, the influence of DNAN is presented in the next section.

#### 4.2.2. Binder Content Influence

It is assumed that an increase in binder content will have a negative effect on the burning velocity of pyrotechnic compositions [1,11,15]. It appears in Figure 4 that despite being an energetic molecule, DNAN plays a role in this way. Without considering magnesium granulometric class, compositions containing 10 wt.% of DNAN lead to lower burning velocities than compositions containing 5 wt.% of DNAN.

Figure 5 presents the quantification (in percentage) of burning velocity increase when reducing binder content from 10 wt.% to 5 wt.%. Results are presented as functions of wt.% Mg.

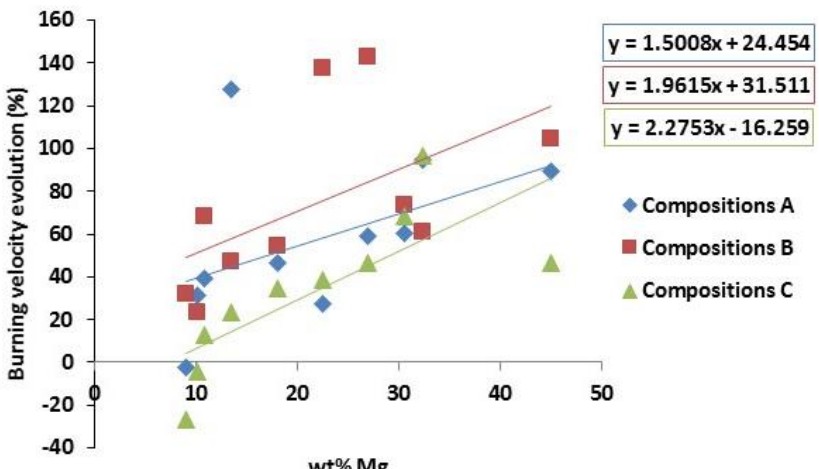

**Figure 5.** Quantification of burning velocity evolution of ternary compositions when reducing binder content from 10 wt.% to 5 wt.%.

The decrease in binder content has a linear influence on the increase of burning velocity for each type of composition (A, B, C). The deviations from linear influence can be explained by great disparities between the corresponding compositions, e.g., A5 and A15. As stoichiometry is supposed to be around 13 wt.% Mg, it appears to be a pivot point in the slope of the burning velocity evolution, especially for composition C. Two linear influences can be seen but with different slopes before and after the stoichiometry for compositions A and C.

With the increase of Mg content, reducing the binder content from 10 wt.% to 5 wt.% tends to increase the burning velocity by a factor of approximately 2. The variations observed can be linked to the change of magnesium granulometry. Specifically, that the decrease of DNAN content has a more important influence on burning velocity with large magnesium particles than with smaller magnesium particles.

### 4.2.3. Magnesium Granulometry Influence

Figure 6 presents the burning velocities measured for compositions A, B, and C at same binder content: 5 wt.% (a) and 10 wt.% (b).

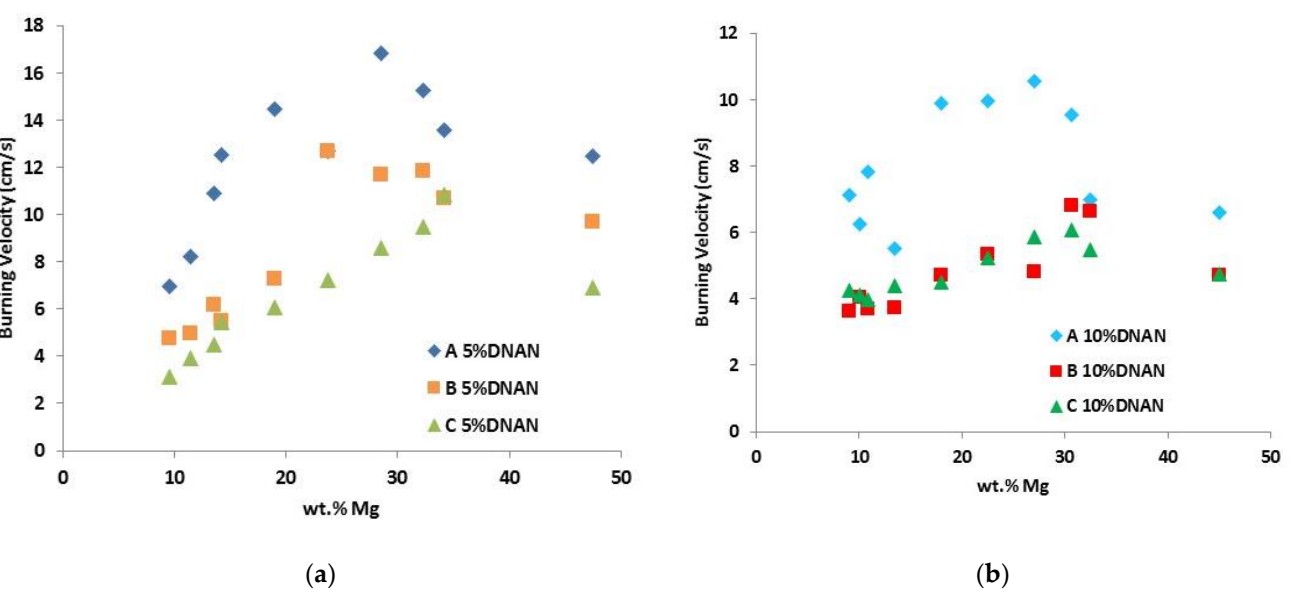

(**a**)          (**b**)

**Figure 6.** Influence of magnesium particle size on the burning velocity of ternary compositions containing 5 wt.% of DNAN (**a**) and 10 wt.% of DNAN (**b**).

As it can be seen in the present work and in other research [1,5], a decrease in Mg particle diameter implies an increase of burning velocity. It has been stated above that maximum burning velocities were not obtained for the same magnesium content when changing granulometric class or binder content.

Granulometric analyses performed on the magnesium classes (A, B, C) have shown a $D_{50}$ close to the lowest diameter of each class, respectively, 140 µm, 200 µm, and 250 µm [9]. Moving from one class to another, C ➜ B; C ➜ A; and B ➜ A, is made by the following diameter evolution:

    C ➜ B: 20% diameter diminution
    C ➜ A: 44% diameter diminution
    B ➜ A: 30% diameter diminution

Figure 7 presents how moving from one Mg granular class to another affects the evolution of the burning velocity (C ➜ B; C ➜ A; and B ➜ A), for the two binder contents studied, 5 wt.% DNAN (a) and 10 wt.% DNAN (b).

When granulometry is reduced from class C to class B (C ➜ B) and for both binder contents, a rather dispersed but constant tendency can be observed on the evolution of the burning velocity. This burning velocity increase is around 30% for compositions containing 5 wt.% DNAN and around 0% for compositions containing 10 wt.% DNAN.

As the burning velocities measured for compositions C and B at 10 wt.% DNAN were extremely close, it can be seen on Figure 6b that the evolution of the burning velocities by switching from Mg granulometric class B and C to A (C ➜ A and B ➜ A) goes accordingly. Moreover, the burning velocity evolution obtained by switching from classes B and C

to granular class A is not monotonic anymore. A maximum can be observed around 20 wt.% Mg content. Burning velocity evolution then decreases beyond 25 wt.% Mg.

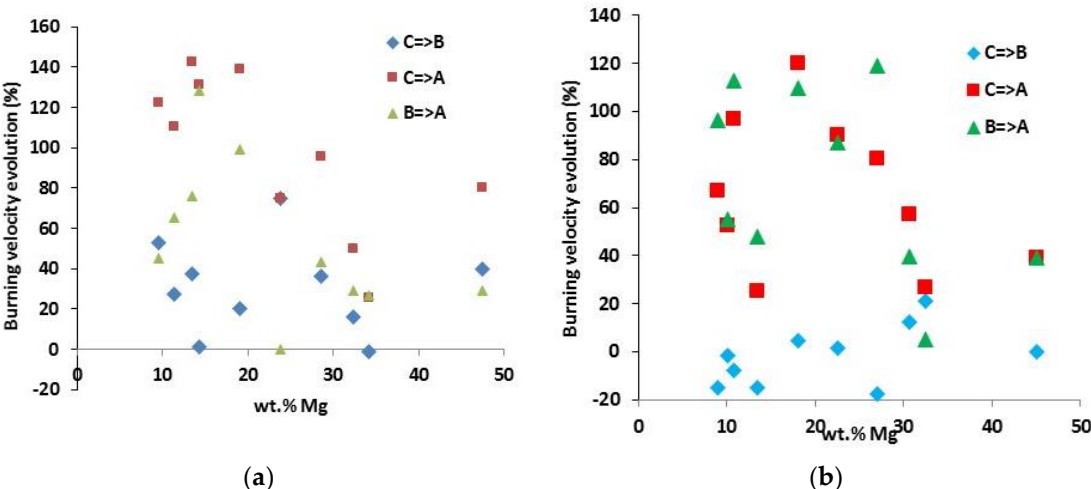

**Figure 7.** Quantification of burning velocity evolution of ternary compositions when changing Mg granulometric class: compositions containing 5 wt.% of DNAN (**a**) and compositions containing 10 wt.% of DNAN (**b**).

Finally, it is outlined that beyond a decrease in particle diameter of about 20%, the influence on burning velocity is not linear anymore. Furthermore, there is a maximum of burning velocity increase obtained for a specific metal reducer content linked to this particle diameter decrease. Indeed, for reducer content higher than 25%, decreasing particle size led to a decrease of burning velocity. We can also notice that this value of 20% is slightly higher than stoichiometry (13%), and maximal burning velocities are generally obtained for reducer content a bit higher than stoichiometry.

## 5. Conclusions

Combustion velocities of binary and ternary pyrotechnic compositions were measured using a standardised experimental method and a modified acquisition setup.

The influence of three granulometric classes of metal reducer was studied, as well as two binder contents and ten ox/red ratios for ternary compositions. It is important to note that the binder contents used in the present work are higher than what it usually used in the industrial field, in order to highlight the binder influence on the burning velocity.

It has been demonstrated that the binder content has a negative influence on the burning velocity of pyrotechnic compositions, despite being an energetic binder, in the present case.

The joint influence of ox/red ratio, binder content and magnesium granulometry has been studied and showed an increase of burning velocity with a decrease of metallic particle diameter as expected. It has also been shown a modification of the nature of the influence of changing Mg granulometric class on the burning velocity. Indeed, beyond a reduction in particle size of 20%, the influence on the burning velocity is not linear anymore. These conclusions are valid for the two binder contents studied, 5 wt.% and 10 wt.%.

Further work will focus on the influence of porosity alone and in addition with the studied influent parameters of the present work, as it is of great influence on the burning velocity [16–19] in order to quantify the joint influence of multiple parameters on the burning velocity of pyrotechnic compositions.

**Author Contributions:** Conceptualization, L.C., P.G. and C.B.; Data curation, C.R.; Investigation, C.R. and L.C.; Methodology, L.C. and P.G.; Project administration, P.G. and C.B.; Supervision, P.G.; Validation, P.G.; Writing—original draft, C.R. and L.C. All authors have read and agreed to the published version of the manuscript.

**Funding:** This research was funded by ANRT and Nexter Munitions and sponsored by CIFRE grant. Grant number: CIFRE No. 2017/1272. They are gratefully acknowledged.

**Institutional Review Board Statement:** Not applicable.

**Informed Consent Statement:** Not applicable.

**Data Availability Statement:** Not applicable.

**Conflicts of Interest:** The authors declare no conflict of interest.

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
