# Peer review of "Burning Velocities of Pyrotechnic Compositions: Effects of Composition and Granulometry"

_energies, doi:10.3390/en15113942_

Round 1

Reviewer 1 Report

This work investigated the effect of powder granulometry and composition on the burning velocities of pyrotechnic composition. Comparing to explosives or propellants, very few studies focused on model pyrotechnic compositions. Therefore, the results provided by the work is valuable. However, the introduction needs to be improved to have more literature summary, such as: 1. Line 25, please provide some literature about the studies of the burning velocity of explosives and propellants? 2. Line 28, “Some of these parameters have been studied in the literature: powder granulometry or composition of the couple oxidant-reducer.” Please give more summary and discussion of the previous work and address the detail of the novelty of present work.

 Additional comments:

Please provide one figure containing raw data obtained from the experimental setup to present the burning velocity calculation process as described in line 107-109.Remove some inside horizontal borders in Table 3.

The paper provides valuable information about the burning behaviors of pyrotechnic compositions, focusing on the effect of mixing ratio and granulometry on burning velocity. It is well written and presented and it is suggested to be published. 

The reviewer has two questions: 

1. line 57-61, it is mentioned that the powder has three granular classes; and line 202, it is mentioned again that the D50 close to the lowest diameter of each class, respectively 140 μm, 200 μm and 250 μm. Is the class C (between 200 and 300 μm) shown in Line 61 correct?

2. Is it possible to give an explanation of the statement of Line 224 " A maximum can be observed around 20 wt.% Mg content. Burning velocity evolution then decreases beyond 25 wt.% Mg"?

Reviewer 2 Report

This paper is well written with engineering significance. The authors should address the following comments.

  1. All the figure quality must be improved.
  2. Simple simulation such as 0D or 1D cases should be conducted to verify the reliability of the measurements.
  3. The research contents alone is too short for a full-length article. The authors should add more measurements, or add more analysis.

Reviewer 3 Report

  1. What is the main question addressed by the research?

The study focuses on the determination of the effects of composition and granulometry of pyrotechnic compositions on burning velocities.

  1. Do you consider the topic original or relevant to the field? Does it
    address a specific gap in the field?

In my opinion, this paper is very topical. I agree with the first sentence of the abstract

“Burning velocities of binary and ternary pyrotechnic compositions have been measured

in gutter”.

  1. What does it add to the subject area compared with other published
    material?

The obvious advantage of this paper is a set of the experimental data which reflect the joint influence of parameters joint influence of several parameters such as oxidant/reducer ratio, reducer granulometry and binder content.  This factor of ‘joint influence’ is absent in the recent paper which is related to similar problems

“Experiments with Pyrotechnic Compositions Based on a Mathematical Model—Part I Evaluation of the Applicability of Mathematical Models in Developing Pyrotechnic Compositions Producing an Acoustic Effect” by Jolanta Bieganska 1 and Krzysztof Baranski

Energies 2021, 14, 8548

  1. What specific improvements should the authors consider regarding the
    methodology? What further controls should be considered?

In my understanding, in the paper statistical aspects are completely absent.

No statistical plan of the experiment based on some mathematical model.

There is no assumption on the error distribution.

There is no consideration on methods of determining the parameters.

No metioning. what is the best method for determining the parameters, the mean-least square method, or maximum likelihood method, or Bayesian approach etc.

The absence of statistical aspects is the main drawback of the paper.

I think, authors should present a clear explanation why the paper neglects statistical aspects. If such explanation will be presented, the paper needs just a minor revision. Otherwise, the paper must be seriously corrected (the major revision).

  1. Are the conclusions consistent with the evidence and arguments
    presented and do they address the main question posed?

The conclusions should somehow touch statistical aspects, i.e., explaining their absence/presence

  1. Are the references appropriate?

            I recommend referring the paper mentioned

“Experiments with Pyrotechnic Compositions Based on a Mathematical Model—Part I Evaluation of the Applicability of Mathematical Models in Developing Pyrotechnic Compositions Producing an Acoustic Effect” by Jolanta Bieganska 1 and Krzysztof Baranski

Energies 2021, 14, 8548

  1. Please include any additional comments on the tables and figures.

Rigorously speaking, the data on tables and figures should be supplemented with a special Appendix with the corresponding statistical analysis.

Round 2

Reviewer 2 Report

The authors do not address the concerns of the reviewer properly. For example, for R2, combustion measurements for chemical mechanisms are verified by 0D or 1D simulations in the combustion community. If no existing models are available, the authors should explain and provide more info. For R3, I agree that the measurements are really diverse in terms of different parameters involved, but no further verification or analysis are provided, making the paper like a experimental report for measured data. I thus requre to see further response from the authors.

Reviewer 3 Report

In this paper, i did not find the statistical analysis of data and obtained relationships. 

Authors must say explicitly about such situation..

Somethin like this

"For this case, the non- statistical analysis of trends is only possible".

I am repeating it must be said explicitly. 

Round 3

Reviewer 2 Report

Although I do not agree with the authors, the results in this paper is very useful in this community. I have no further comments.